# Fiber Laser Welded Cobalt Super Alloy L605: Optimization of Weldability Characteristics

**DOI:** 10.3390/ma15217708

**Published:** 2022-11-02

**Authors:** B. Hari Prasad, G. Madhusudhan Reddy, Alok Kumar Das, Konda Gokuldoss Prashanth

**Affiliations:** 1Defence Research & Development Laboratory, Hyderabad 500058, India; 2Defence Metallurgical Research Laboratory, Hyderabad 500058, India; 3Department of Mechanical Engineering, Indian Institute of Technology (ISM), Dhanbad 826004, India; 4Department of Mechanical and Industrial Engineering, Tallinn University of Technology, 19086 Tallinn, Estonia; 5Erich Schmid Institute of Materials Science, Austrian Academy of Sciences, Jahnstrasse 12, 8700 Leoben, Austria; 6CBCMT, School of Mechanical Engineering, Vellore Institute of Technology, Vellore 630014, India

**Keywords:** fiber laser welding, microstructure, residual stress, epitaxial growth, fractography, EBSD analysis

## Abstract

The present study describes the laser welding of Co-based superalloy L605 (52Co-20Cr-10Ni-15W) equivalent to Haynes-25 or Stellite-25. The influence of laser welding process input parameters such as laser beam power and welding speed on mechanical and metallurgical properties of weld joints were investigated. Epitaxial grain growth and dendritic structures were visible in the weld zone. The phase analysis results indicate the formation of hard phases like CrFeNi, CoC, FeNi, and CFe in the weld zone. These hard phases are responsible for the increase in microhardness up to 321 HV_0.1_ in the weld zone, which is very close to the microhardness of the parent material. From the tensile strength tests, the ductile failure of welded specimens was confirmed due to the presence of dimples, inter-granular cleavage, and micro voids in the fracture zone. The maximum tensile residual stress along the weld line is 450 MPa, whereas the maximum compressive residual stress across the weld line is 500 MPa. On successful application of Response Surface methodology (RSM), laser power of 1448.5 W and welding speed of 600 mm/min i.e., line energy or heat input equal to 144 J/mm, were found to be optimum values for getting sound weld joint properties. The EBSD analysis reveals the elongated grain growth in the weld pool and very narrow grain growth in the heat-affected zone.

## 1. Introduction

Cobalt-based superalloy L605 is a preferred material for a wide range of high-temperature applications such as gas turbine blades, jet engine parts, aerospace systems, combustion chambers, etc. Owing to their excellent mechanical and fatigue strength; biocompatibility; corrosion and oxidation resistance, they also have wide applications in chemical, marine, and bio-medical industries [1,2,3]. The presence of cobalt and chromium makes it eligible for extensive biomedical applications. The presence of tungsten, chromium, and molybdenum enhances the melting point, hardness, and density of the said alloy [4]. Co-based superalloys are synthesized through a vacuum induction melting process followed by electro slag refining, which leads to very low non-metallic inclusions and hence limited crystal defects. Sometimes, it is also observed that this alloy is superior to Ni-based superalloys as it has better thermal shock resistance and anti-corrosion properties even in a hot gas environment. Different welding processes with high metal deposition rate like Gas Metal Arc Welding (GMAW), Gas Tungsten Arc Welding (GTAW) showed wide HAZ and welding defects in the joints which restricts its uses in precision aerospace components [5]. Laser welding produces minimum defects and narrow HAZ, which attracts its applications in various industries. Although there are several laser welding processes (using gas lasers and solid-state lasers), fiber laser welding is preferred over other processes due to its low maintenance cost and high reliability of the laser source.

Welding of different superalloys has been reported by many researchers. For example, Osoba et al. [6] have carried out laser welding of Haynes 282 superalloy and observed the formation of micro-segregation pattern in the fusion zone. Chen et al. [7] have investigated the grain growth phenomenon in Co-Cr-Mo alloy during laser melting and reported that the solidification starts from the epitaxial grains in the boundary zone of the melt track. The planar grain growth is not possible due to very high grain growth velocity [8]. M. Shamanian et al. [9] carried out pulsed Nd: YAG laser welding of Co-based superalloys and observed that heat input to the welding process plays a major role in the control of microstructure and grain orientation in the weldment. However, the changes in heat input had an insignificant effect on the mechanical properties. Palanivel R. et al. [10] studied the Nd: YAG laser welding of IN 800 and observed the formation of an elongated columnar but fine equiaxed dendritic structure in the fusion zone. In addition, the phase transformation occurred due to the higher cooling rate of the laser welding process. The ductile failure of welded joints was observed at higher welding speed whereas brittle failure occurred at low welding speed. Similarly, many kinds of literature describe the effects of input parameters of the laser welding process on the weld quality parameters of different materials such as the microstructure and grain orientation [11,12,13,14], bead profile [15], mechanical properties [16,17,18], metallurgical properties [19], corrosion resistance [20] and residual stresses [21].

In all the above-mentioned applications, the joining plays a crucial role. Some applications demand better weld geometry with the narrow heat-affected zone, which is difficult to meet through conventional welding processes [22,23,24,25,26]. These requirements may be met using a highly focused and concentrated heat source, as in laser and electron beam welding processes. The present investigation deals with the fiber laser welding of Co-based superalloy L605 of 2 mm thick sheets and describes the detailed analysis of the welded joints for weld bead geometry, microstructure, microhardness, tensile yield strength (yield stress), phase analysis, residual stress and EBSD study. The statistical analysis of the welding process has been carried out to develop a regression model and to correlate the dependent and independent process parameters.

## 2. Materials and Methods

### 2.1. Setup Configuration

The fiber laser welding setup consists of a 4 kW fiber laser source (make: Arnold Ravensburg, Germany), X-Y-Z CNC stages (movement range: 2500 mm × 3000 mm × 750 mm, maximum traverse speed in X-Y plane: 6000 mm/min) and an integrated CNC controller. The emission wavelength of the fiber laser is 1070 nm; the spot diameter on the focal plane is 440 µm; the beam profile is Gaussian and can be operated in continuous wave mode. The laser beam is delivered onto the worktable through an optical fiber cable. The core diameter of the optical transmission fiber is 200 µm. The welding head is equipped with an air cross jet facility to protect the lens from the deposition of metal vapors. For shielding the weld zone from atmospheric contamination, inert gas (argon) is purged from the top and bottom (above and below the workpiece) of the weld zone. The welding head is provided with X-Y-Z motion; whereas the workpiece is mounted on the machine table using a rigid fixture. The setup configuration for conducting fiber laser welding experiments is presented in Figure 1.

### 2.2. Workpiece Preparation

Cobalt-based superalloy L605 sheets having a thickness of 2 mm were used as workpieces to carry out the experiments. The samples were prepared to 60 mm × 125 mm × 2 mm size by abrasive water jet cutting, followed by milling and fine grinding of the edges. The prepared samples were cleaned with acetone to remove the oxide layer from the surfaces and the edges before conducting the welding experiments. The chemical composition of L605 material is given in Table 1.

### 2.3. Parameter Selection

Several pilot experiments were conducted by varying laser parameters such as laser beam power and welding speed at the focal plane (Figure 2) with reference to the workpiece surface for finding the ranges of different parameters to ensure full penetration welding along with the formation of acceptable bead geometry. Subsequently, the ranges of different weld parameters were selected as laser beam power: 1400–1800 W, laser beam scanning speed: 200–600 mm/min.

### 2.4. Design of Experiments

In the present research, Response Surface Methodology (RSM) was followed for conducting weld experiments. Table 2 shows the levels of different input welding parameters which were decided after conducting trial experiments. Central Composite Design (CCD) was used to find the parameters setting of each experimental run. MINITAB software was used for conducting Design of Experiments (DOE). Table 3 shows CCD with parameter settings for 14 experimental runs. RSM is a statistical approach to observe the influence of simultaneous variation of any two input parameters on the responses which cannot be done in a normal parametric study, and it provides the designed model (Equation (1)) which can predict the responses with the variations of input welding parameters within the selected range.
Y = α_0_ + α_1_ X_1_ + α_2_ X_2_ + α_3_ X_3_ + α_12_ X_12_ + α_13_ X_13_ + α_23_ X_23_ + α_11_ X_11_ + α_22_ X_22_ + α_33_ X_33_(1)
where, *Y* represents the response variable, *X*_1_, *X*_2_, and *X_3_* are the input process variables, α_0_ is a constant, *α*_1_, *α*_2_, and *α*_3_ are linear coefficients, α_11_, *α*_22_, and *α*_33_ are quadratic coefficients, and α_12_, α_13_, and α_23_ are interaction coefficients. The prediction ability of the developed equation is checked through ANOVA.

### 2.5. Experimentation

The prepared samples (size: 60 mm × 125 mm × 2 mm) were mounted on the welding fixture for making a butt joint without using filler material (autogenous welding). All the experiments were conducted using the parameters settings listed in Table 3. The laser source was operated in continuous mode, with a focus spot size of 0.44 mm at the focal plane. During the experimentation, the laser was irradiated perpendicular to the work surface during the welding experiments. Commercial argon gas (purity > 95%) was used as the shielding gas and purged at the rate of 10 L/min at 1 bar pressure. The gas purging was done from the top and bottom of the weld zone to prevent oxidation and other atmospheric contaminations. In this way, 14 welded samples of L605 sheets were prepared and subjected to quality evaluation. The fabricated samples were cut from the weld specimens using a wire-EDM machine and molds were prepared for mounting purposes followed by polishing to measure the microhardness.

### 2.6. Characterization Processes

At first, all-welded samples underwent a radiography test using a 160 kV X-ray source (model: Seram 235; make: BALTAUE NDT, Canada) to identify the internal defects (such as porosity, and internal cracks) present in the weld zone, which is required before conducting the tensile strength test. A dye penetrant test was conducted to detect surface defects. Then the tensile test specimens (prepared as per ASTM E8/E8M-15a standard) were cut from minimum defects or defect-free zones of weld samples using the wire-EDM process (Figure 3a,b). The tensile tests were carried out on Universal Tensile Testing Machine (UTM) (model: BiSS make Measure India Corporation Pvt Ltd., Secunderabad, India). The remaining part of the welded joint was kept for evaluation of other properties. The bead geometry of the weld bead was analyzed using a metallurgical microscope (make Olympus Corporation, Tokyo, Japan). The different parameters of the weld bead geometry are presented in Figure 3c. The samples were polished and etched with a chemical solution (20 mL HCl, 5 mL HNO_3_, 65 gm FeCl_3_, 150 mL distilled water, swabbing for 10–15 s) for measuring the heat-affected zone (HAZ) and microstructure analysis using Metallurgical Microscope (model: BX51M; make: Olympus Corporation, Tokyo, Japan). Microhardness tests were performed on the weld zone and on the parent material for comparison using a Digital Microhardness tester (model: MMT-X7; make MATSUZAWA, Akita, Japan). Phase and chemical composition analysis was carried out by Scanning Electron Microscope (SEM) (model EVO MA10; make ZEISS, Jena, Germany). XRD (make: Rigaku, Tokyo, Japan) analysis was done to identify the new chemical compounds formed due to welding temperature. Residual stress measurement on weld samples was carried by LXRD equipment (make: Proto, Canada). EBSD study was carried out for the welded samples in the weld zone to analyze the grain growth by FESEM with an attachment of EBSD Camera (Make: ZEISS, Jena, Germany); the results of the characterization process were analyzed and discussed in Section 3.

## 3. Results and Discussion

All the prepared weld samples (14 numbers) were subjected to different tests as discussed in Section 2.6 to evaluate the characteristics of weld joints. The prominent test results and the discussions are enumerated in the following sections.

### 3.1. Radiography and Dye Penetrant Test

The weld zones were subjected to radiography testing as per ASTM E1742 with X-rays to find any internal defects present within them. Similarly, the dye penetrant test was carried out as per ASTM E1417 to check the surface defects. Few of the captured images of radiography and dye penetrant test are presented in Figure 4. In the radiography test, the weld samples were kept above the phosphoric image plate and then high-power X-rays (160 kV, 4 mA, exposure time: 2 min) were made to fall on the welded samples. A latent image of the sample was captured by the plate. These latent images were converted into real viewable images by a laser scanner (make GE, USA) and then images were further analyzed for internal defects. In the dye penetrant test, the liquid penetrant (Magna flux SKL-SP1) was spread over the weld zone after a thorough cleaning by the cleaner (Magna flux SKC-1) and left untouched for 30 min. Subsequently, the developer (Magna flux SKD-S2) is spread over the weld bead and left for 5 min, which pulls up the penetrant and gets accumulated in the pores and cracks. The surface turns red by which the surface defects are identified [27,28]. The results of the radiography test and die penetration tests conducted on the samples are shown in Figure 4. The presented figures show that samples are free from embedded pores and internal cracks. Only a few undercuts are visible at the edge of the welded specimen that are not the part of samples used for various tests. Tensile test specimens were cut from the weld coupons, which were having minimum or zero defects.

### 3.2. Statistical Analysis

After the radiographic test, the other quantitative responses were measured (Table 3) and entered into the design table of RSM with CCD. All the data were analyzed to observe the effects of the combination of different process parameters on the obtained responses. Regression equations in uncoded form (Equations (2)–(6)) were developed and validated through the ANOVA (Analysis of Variance) (Table 4) which are described below. Figure 5 represents the normal probability plots for different responses. As the distribution of different points is very close to the straight line, which indicates the model has good prediction ability [29,30]. The values of R-sq and R-sq(adj) are very close to each other, which indicates the less variability of the predicted responses with respect to the input parameters. The P values in the ANOVA table for linear, square, and interaction of input parameters are found to be less than 0.05 which indicates that the developed regression model is significant [31].
W_1_ (µm) = −16422 + 28.75 P − 15.30 WS − 0.00807 P^2^ + 0.01387 WS^2^ − 0.00174 P·WS(2)
W_2_ (µm) = −34885 + 54.1 P − 23.26 WS − 0.01718 P^2^ + 0.00709 WS^2^ + 0.00676 P·WS(3)
H_1_ (µm) = 680 − 1.45 P + 2.55 WS + 0.000893 P^2^ + 0.000805 WS^2^ − 0.002470 P·WS(4)
H_2_ (µm) = −506 − 1.54 P + 7.04 WS + 0.00162 P^2^ + 0.00291 WS^2^ − 0.006685 P·WS(5)
Y (MPa) = −2537 + 4.44 P − 1.864 WS − 0.001580 P^2^ + 0.000715 WS^2^ + 0.000951 P·WS(6)

#### 3.2.1. Weld Bead Geometry Analysis

The weld bead geometry plays a vital role in the better strength of the joint. One of the typical weld bead geometries is shown in Figure 3c for reference purpose and the measured values are presented in Table 3. Equations (2)–(6) represents the variation of geometrical parameters (responses) corresponding to the inputs which are explained with the help of the contour plots in Figure 6. The width of the weld zone is reduced with the increase in scanning speed, this might be due to laser beam passes quickly over the weld line and hence less amount of base metal is melted, however, the weld width increases reasonably with the increase in laser power. With the increase in laser power, the bead width increases due to over melting of the base material (Figure 6b). The bead width (W2) at lower side of the welding is found to be reduced with the increase in scanning speed, it may be due to the formation of keyhole in the welding process which leads to wider top and narrow bottom of the weld bead (Figure 6c). The bead height and undercut are found to be maximum at higher laser power and lower scanning speed, at the appropriate combination of laser power 1600 W and scanning speed of ~500 mm/min the bead height and undercut are found to be minimum as shown in Figure 6d,e. It shows the variation of W_1_, W_2_, H_1_, and H_2_ with respect to the laser beam power and welding speed. It has been observed that the minimum weld bead geometry (Table 3) with top width (W_1_): 2336.85 µm, Bottom width (W_2_): 1790.43 µm height (H_1_): 104.64 µm and undercut (H_2_): 0 µm were obtained at different parameter setting of laser power in the range of 1400 W to 1800 W and scanning speed 200 to 600 mm/min and the same is also confirmed from the contour plots shown in Figure 6. The optimum values of the bead geometry can be obtained at parameters settings of laser power 1448.5 W and scanning speed 600 mm/min where top width (W_1_): 2594.45 µm, bottom width (W_2_): 1848.62 µm, height (H_1_): 126.04 µm and undercut (H_2_): 130.14 µm. In view of the above, the welding parameters can be selected from the contour plots shown in Figure 6 as per the requirement of the weld bead geometry, which is in line with the reference article [32].

#### 3.2.2. Microhardness of Weld Bead

The prepared sample for microhardness is presented in Figure 7a. The micro-hardness was measured from the centerline of the weld bead to one side towards the parent material. Figure 7b represents the microhardness profile of a typical weld sample. From the experimental data, it is observed that there is little variation in microhardness (maximum value: 321 HV_0.1_) on the weld bead as compared to the parent material (average value: 305 HV_0.1_) that indicates a good quality of weld joint without impacting its mechanical properties. During the laser welding, when the laser beam is irradiated on the material, considerably high temperature is developed and melts the weld sample interface. When the material solidifies, the fusion zone is created. Some parts of the heat are dissipated into the welding parts and lead to the formation of HAZ. Due to very short exposure time of the laser, the temperature of the weld zone decreases rapidly with high cooling rates. It results in the formation of austenite and austenite + carbide phases along with other different phases like CrFeNi, CoC, FeNi, and FeC that affect the microstructures of the fusion zone and microhardness is increased.

#### 3.2.3. Tensile Yield Strength (Yield Stress)

To perform the tensile strength tests, the specimens were prepared using the wire-EDM process which is in accordance with ASTM E8 standard (Figure 3a), and the prepared tensile test specimens are shown in Figure 8a. The mounting of the specimen on UTM is shown in Figure 8c. The tensile tests for all specimen along with parent material were conducted at room temperature with a crosshead speed of 0.5 mm/min and the captured plots are shown in Figure 8d,e. The plots indicate that the amount of elongation varies between 5.21% to 25.09%, whereas the breaking load varies from 1 kN to 12 kN. During the tensile strength tests, all the samples failed in the weld zone as shown in Figure 8b and this is because of epitaxial grain formation, which was located along the axis of the weld bead [33]. The solidification of molten metal takes place at room temperature and the parent material on both sides of the weld zone acts as the chiller unit due to which the solidification starts from both sides, as a result, the end of solidification takes place at the center of the weld nugget [34]. During the solidification process, the equiaxed grain structure is formed in the weld zone due to which a parting line is observed at the center of the weld zone and that leads to the formation of the weaker part. Therefore, the optimization of the input process parameters was performed for better strength of the weld joint. It was also observed that as the heat input supply during the welding is reduced, the cooling rate increases, and it results in the formation of finer grain size and therefore results in enhanced mechanical properties [35]. In this way, laser welding parameters at high input i.e., 1800 W laser power and 200 mm/min scanning speed resulted in low yield stress of 299 MPa as compared to the laser welding parameters at low heat input i.e., 1400 W laser power and 600 mm/min, where yield stress of 546.20 MPa was obtained.

The measured values of tensile yield strength were fed to the design table and further analyzed to correlate with the input parameters presented in Equation (6). The contour plots (Figure 6a) explain the variation of yield stress w.r.t input parameters and it is observed that the tensile yield strength (yield stress) up to 541 MPa can be obtained at a parameter combination of laser power: 1448.5 W, scanning speed: 600 mm/min i.e., line energy or heat input equal to 144.44 J/mm, as shown in Figure 9.

#### 3.2.4. Optimization of Process Parameters

In the present work, the multi-objective optimization of input process parameters was carried out using the desirability approach through the RSM technique considering responses: W_1_, W_2_, H_1_, H_2_ and Y. The desirability values of response parameters are shown in Figure 9. It is observed that the desirability value approaches 1 which indicates the good prediction ability of the developed model. In the present study, laser beam power: 1448.5W, scanning speed: 600 mm/min give response values such as W_1_: 2594.45 µm, W_2_: 1848.62 µm, H_1_: 126.04 µm, and H_2_: 130.14 µm and Y: 546.20 MPa (Figure 9). These responses were validated by conducting the confirmation experiments and the error was found to be within 5%.

### 3.3. Analysis of Weld Bead

The mechanical and metallurgical properties were evaluated further to investigate the weldability characteristics of L605 alloy. The prepared weld samples which had high and low heat input were considered in this investigation, which are presented in the following sections.

#### 3.3.1. Morphology and Microstructure

The prepared weld samples were polished to get a mirror finish surface and etched with a chemical solution to observe the microstructure by metallurgical microscope and SEM (Figure 10). Figure 11c,d represents the morphology of the cross-section of the weld bead. Small concavity is observed in the weld root as shown in Figure 10c, which may be due to high heat input during the welding process with parameters laser beam power 1800 W, scanning speed 200 mm/min. The flat root is observed in the other case as shown in Figure 10d which is due to the low heat input with parameters laser beam power 1400 W, scanning speed 600 mm/min. It is also worth stating that, the transfer of heat within the weld pool happens by convection; and therefore, the fluid dynamics predominantly determine the formation of bead shape. The major factors coming into play are surface tension, volume contraction, vapor pressure, phase transformation and gravity [36]. With the reduction in energy input, the cross-section of the weld joint transforms into almost an X-shape; whereas the V-shaped cross-section is obtained when there is partial keyhole formation [37]. This leads to a reduction in the width of the weld zone, as shown in Figure 10b. Figure 10e,f indicate the magnified images of Figure 10c,d, respectively, which affirm that granular and epitaxial grain growth takes place during the solidification process.

In the present work, the two typical samples fabricated at laser power of 1400 W and 200 mm/min scanning speed, and 1800 W and 200 mm/min were considered for the microstructural analysis through SEM. The detailed investigation of the weld section at the interface and weld zone has been selected. The SEM micrographs revel the long columnar dendritic structure, which is divided into cellular and columnar dendritic structures. Figure 11 and Figure 12 represent microstructure and Energy Dispersive Spectroscopy (EDS) results of the weld bead. From Figure 11, it has been observed that the dendritic microstructures are formed having long columnar grains at the interface (Figure 11b) and cellular dendrites (Figure 11c). The fusion zone microstructure at this section revels formation of secondary dendritic arm, the magnifying image shows the formation of intermetallic phase whose presence is confirmed through the EDS report (Figure 11d), the grains are parallel and distributed uniformly with mix coarse and fine columnar dendritic (uniaxial) which might be due to the proper mixing of material at the fusion zone [38,39]. Figure 12 shows the SEM images of the weld cross section, i.e., at the interface and the fusion zone; this shows the microstructure with no directional solidification with columnar and cellular dendritic structure. The white granular structure confirms the present of Co, which is confirmed through EDS report, as shown in Figure 12d. The Variation in thermal properties like specific heat and thermal conductivity with changing temperature of the base alloys leads to the formation of microstructure obtained after welding. The presence of Co, Cr, W, Ni, and Mn was observed in the inter-dendritic structure at the interface (Figure 11b,d and Figure 12b,d). On the other hand, the laser welding process associated with a rapid cooling rate without pores and cracks was observed at the interface zone in the prepared weld samples (Figure 11a and Figure 12a) [40]. The columnar dendritic and cellular dendritic structures were formed in the weld zone, and it may happen due to the formation of CrFeNi, CoC, FeNi, and CFe phases in the inter-dendritic regions and this provides strength to the weld zone [41].

#### 3.3.2. Phase Analysis

X-ray diffraction (XRD) test was performed with laser parameter settings of 1800 W, 200 mm/min, and 1400 W and 600 mm/min on the weld bead cross-section to examine the presence of different phases (Figure 13a) along with XRD plot of parent material as shown in Figure 13b. The 2θ angle varied from 30 to 90° with a scanning speed of 4 deg/min. Figure 13 shows the XRD spectra of the weld beads and the analysis shows the presence of CrFeNi, CoC, FeNi, CFe, and Fe phases in the weld zone, unlike parent material.

#### 3.3.3. EBSD Analysis

The base metal EBSD micrograph is shown in Figure 14. The image quality map show in Figure 14 indicates a uniform fine grain size. The average grain size of the base metal is found to be ~17 µm. The fine grain size has uniformly distributed grains as shown in the Inverse pole figure map. The microstructure has around 4 % low angle grain boundaries with the remaining 96% high angle grain boundaries, as shown in Figure 14c. The pole figures and inverse pole figures show a random orientation of the grains in the base metal with no specific texture. The microstructure of the L605 weld pool region on both sides of the weld central line is shown in Figure 15. The average grain size of the HAZ has increased to 68 µm from the base metal grain size of 17 µm. This region is melted during the welding, and a dendritic/ columnar grain growth result in the formation of long elongated grains and the direction of the grains are elongated towards the base metal. The fraction of the low angle grain boundaries has increased from 4% to 15%. The resultant increase in LAB is due to the intersection for growing columnar grains. The pole figure and inverse pole figures indicate that there is a strong orientation of grains in the direction of (111) and (101), the resulting texture is like cold rolled texture.

The EBSD maps of the complete weld bead region along with base metal are shown in Figure 16. The microstructure shows a very narrow HAZ, wherein a uniform grain size has been noticed. This region shows the grain growth compared to base metal, as this region is exposed to higher temperature below the melting point of the base material. The strain energy stored in the base material along with the temperature above recrystallization temperature has resulted in grain growth. The image quality map shows a completely dendritic and elongated grains. The pole figures indicate an overall texture orientation in <101> direction. The phase map presented indicates a fully austenitic structure with small amounts of metallic carbides. The carbide formation of (W, Cr)_7_ C_3_ was observed.

#### 3.3.4. Residual Stress Measurement

Residual stress was measured through the XRD technique using LXRD equipment on two prepared weld samples (input parameters 1800 W, 200 mm/min and 1400 W, 600 mm/min) at seven different points along mutually perpendicular directions (D90° and D0°) with respect to the weld line as shown in Figure 17a,b. In Figure 10c,d the grains are elongated across the weld line. It is observed the residual stress is compressive when measured along D90° (Figure 17c,d) for both parameter settings. Along D0° the compressive stress is reduced and sometimes it becomes tensile (Figure 17c) for parameters setting of 1800 W, 200 mm/min. However, at parameters setting of 1400 W, 200 mm/min, give weld bead with tensile residual stress along D0° (Figure 17d). Because of the above, the input parameters which produce the elongated grains may be preferred for the laser welding process.

#### 3.3.5. Fractography Study

Figure 18 shows the SEM micrograph of the fractured surfaces of the tensile test specimens at high heat input parameters of 1800 W, 200 mm/min, and low heat input parameters of 1400 W, 600 mm/min. It is observed that tensile test specimens are subjected to ductile failure as well as cleavage failure, as illustrated in Figure 18. A significant number of dimples were observed in the fracture zone. In the inter-granular cleavage, micro-voids are seen at high and low energy input. The observed features show the ductile failure of the welded samples [42,43].

## 4. Conclusions

Fiber Laser welding of L605 cobalt superalloy sheets of a thickness of 2 mm was carried out successfully and different characterization techniques were followed for evaluation of the properties of the welded joints. From the study, the following conclusions are drawn:Response surface methodology was used for the development of regression equations to correlate the input and output process parameters. Within the present range of laser welding parameters, the work surface was kept at the laser beam focal plane, which gave different weld quality in the experimentation at different laser input parameters.Regression models were developed, and their predictability was checked through ANOVA. The best weld bead geometry of top width (W1): 2594.45 µm, bottom width (W2): 1848.62 µm, bead height (H1): 126.04µm, and undercut (H2): H2: 130.14 µm along with the highest tensile yield strength (yield stress) of 546.20 MPa were obtained at parameters setting of laser beam power: 1448.5W, scanning speed: 600 mm/min.The failure of tensile test samples occurred in the weld zone due to the formation of epitaxial grains, which was confirmed from the optical micrographs and to avoid this wobble mode of fiber laser welding may be adopted. The ductile failure occurred in the weld zones of the tensile test specimens, which are confirmed by the presence of dimples, intergranular cleavage, and micro voids in the fractured surfaces.The SEM analysis indicates the absence of micro-cracks in the weld zone. The microstructure (EBSD analysis shows a narrow HAZ, wherein a uniform grain size has been noticed. This region shows the grain growth compared to base metal, as this region is exposed to higher temperatures below the melting point of the base material. The strain energy stored in the base materials, along with the temperature above recrystallization temperature, has resulted in grain growth.All experiments in this work were conducted at room temperature and the material properties reported here also carried at room temperature. However, testing and characterization at high operating temperature will be considered as a part of future work.

## Figures and Tables

**Figure 1 materials-15-07708-f001:**
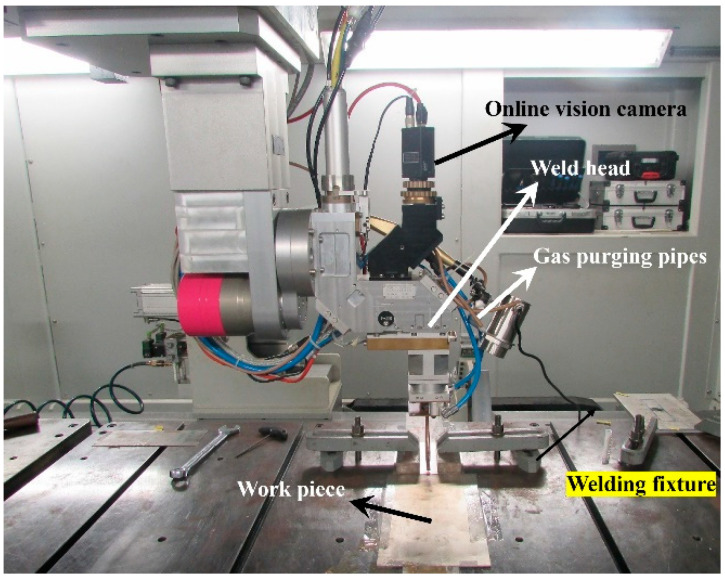
Setup configuration of the fiber laser welding process.

**Figure 2 materials-15-07708-f002:**
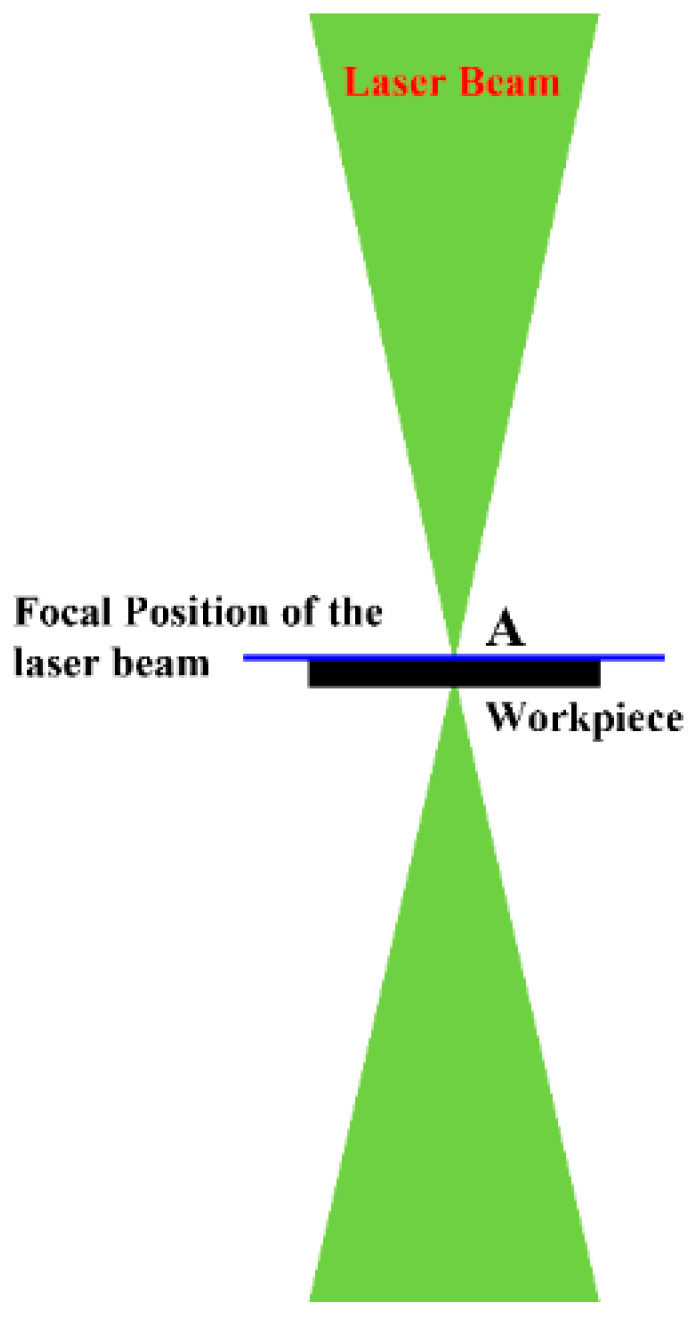
Position of the workpiece surface at the focal position of the laser beam.

**Figure 3 materials-15-07708-f003:**
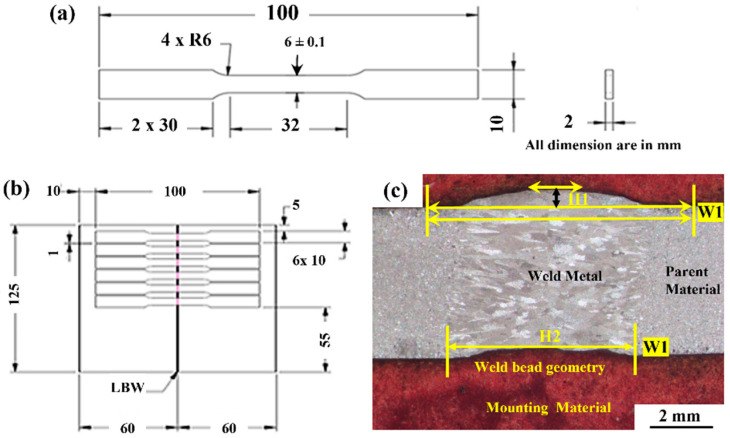
(**a**) Dimensions of the prepared tensile test specimen, (**b**) Tensile test specimens in the weld sample (**c**) Weld bead geometry on prepared weld sample (H1: weld bead height above the workpiece surface, H2: Weld undercut at the bottom surface, W1: Weld bead width at the top surface, W2: Weld bead width at the bottom surface).

**Figure 4 materials-15-07708-f004:**
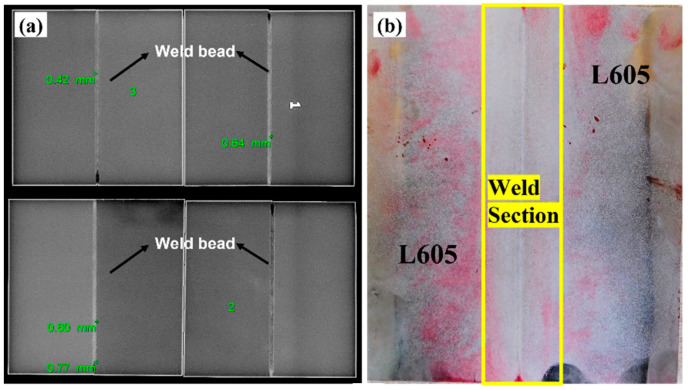
Captured images of (**a**) radiography test and (**b**) dye penetrant test samples.

**Figure 5 materials-15-07708-f005:**
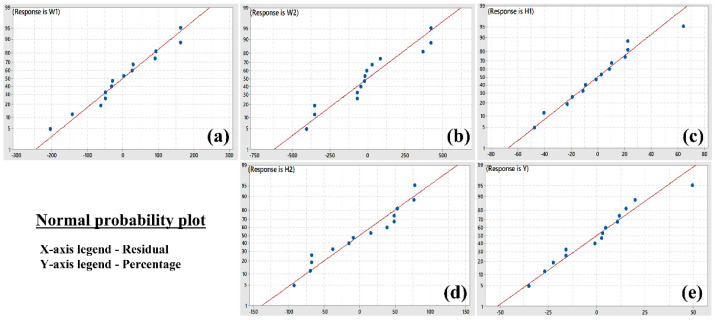
Normal probability plots for responses (**a**) Weld bead width at the top surface (W_1_), (**b**) Weld bead width at the bottom surface (**c**) Weld bead height above the workpiece surface (H_1_), (**d**) Weld undercut at the bottom surface (H_2_), (**e**) Tensile yield strength (Y).

**Figure 6 materials-15-07708-f006:**
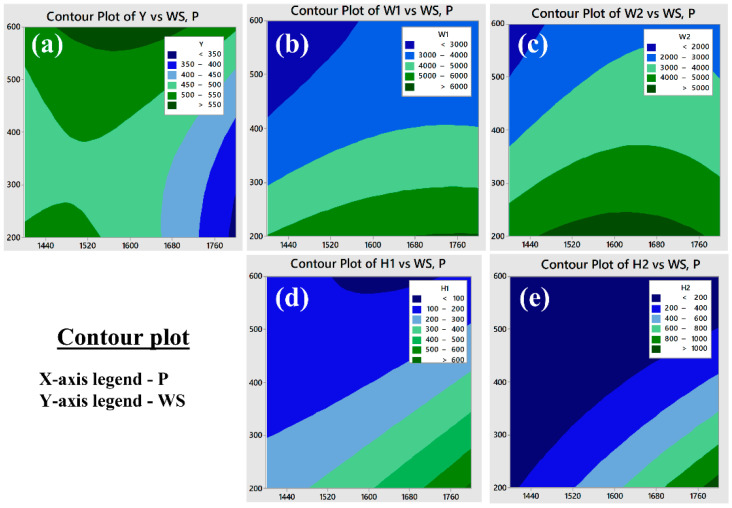
Contour plots for weld bead geometry: (**a**) variation of tensile yield strength (Y) (**b**) variation of top width of weld bead (W_1_), (**c**) variation of bottom width of weld bead (W_1_), (**d**) variation of weld bead height (H_1_), (**e**) variation of undercut (H_2_), w.r.t input parameters.

**Figure 7 materials-15-07708-f007:**
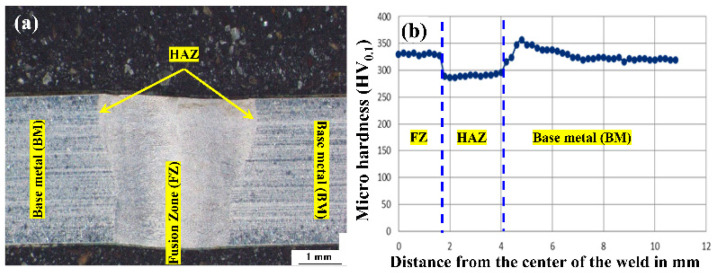
(**a**) Typical weld bead, (**b**) Microhardness plot of typical weld sample.

**Figure 8 materials-15-07708-f008:**
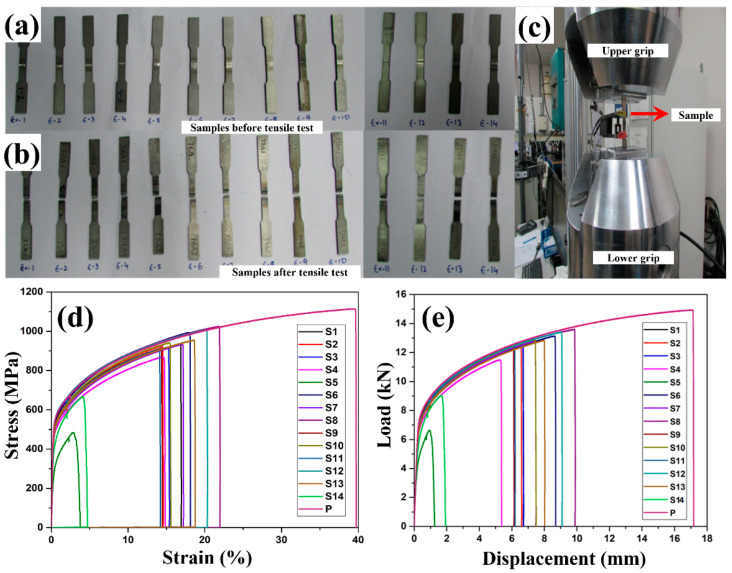
(**a**) Tensile test specimens (**b**) Images of tensile test specimens after failure (**c**) UTM configuration for the tensile test (**d**,**e**) Tensile stress-strain, load-displacement plots 3.7. Wettability studies.

**Figure 9 materials-15-07708-f009:**
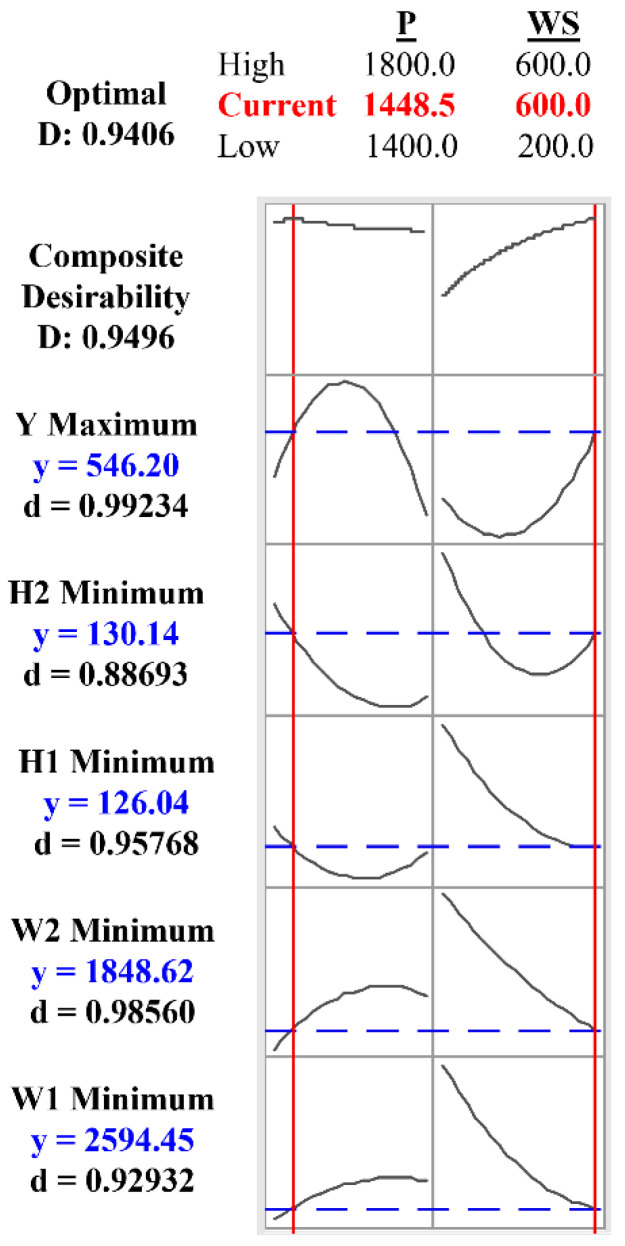
Optimization plot for the weld bead.

**Figure 10 materials-15-07708-f010:**
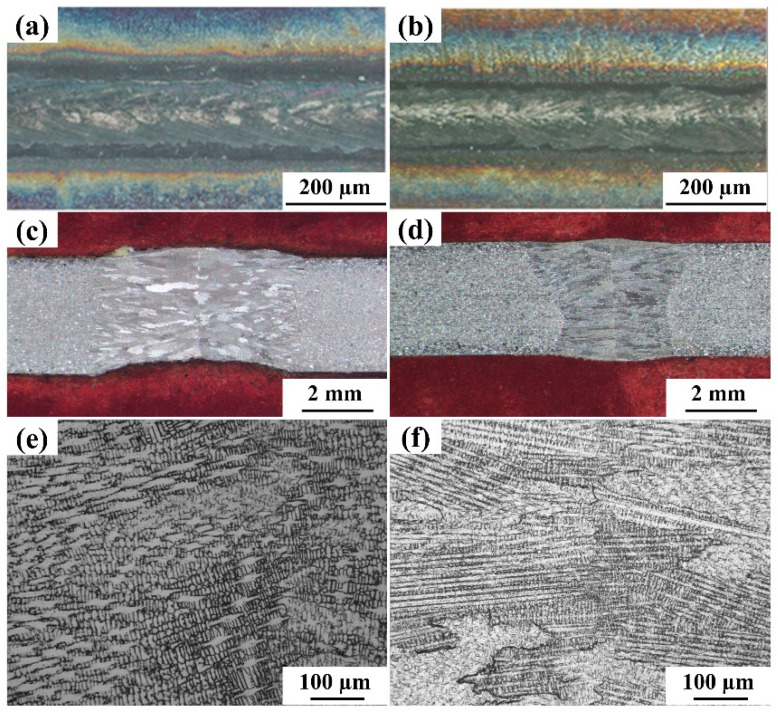
Optical images of surface and cross-section of welded joints (**a**,**c**,**e**) at 1800 W, 200 mm/min and (**b**,**d**,**f**) at 1400 W, 600 mm/min.

**Figure 11 materials-15-07708-f011:**
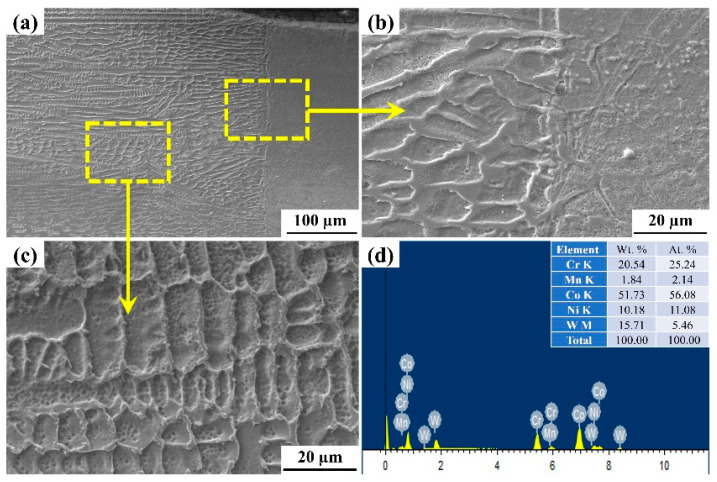
SEM images of the weld joint with high heat input (weld parameters: 1800 W, 200 mm/min). (**a**) Weld cross section (**b**) Interface zone (**c**) Microstructure (**d**) EDS report.

**Figure 12 materials-15-07708-f012:**
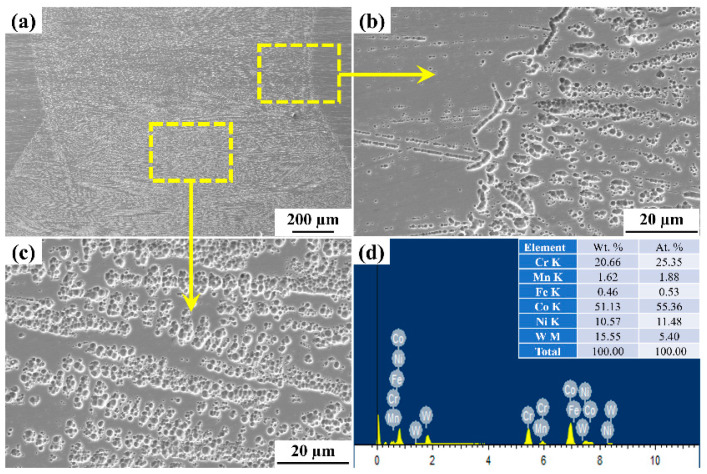
SEM images of welded joints with low heat input (weld parameters: 1400 W, 600 mm/min). (**a**) Weld cross section (**b**) Interface zone (**c**) Microstructure (**d**) EDS report.

**Figure 13 materials-15-07708-f013:**
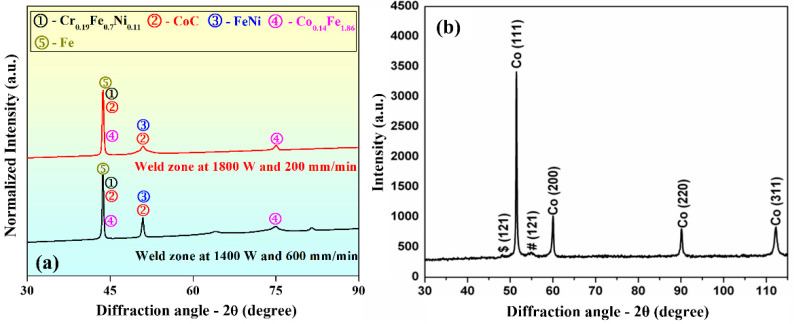
XRD spectra of the (**a**) weld zones processed at 1800 W, 200 mm/min, and 1400 W, 600 mm/min, respectively, and (**b**) parent material.

**Figure 14 materials-15-07708-f014:**
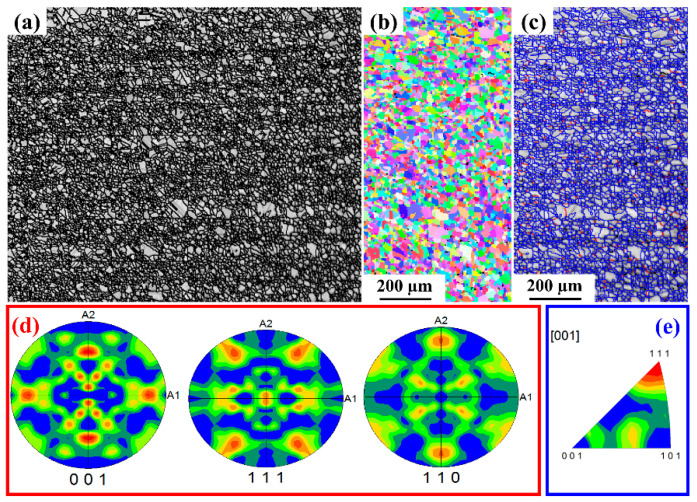
Base metal microstructure (**a**) Image quality map (**b**) Inverse pole figure color map (**c**) LAGB and HAGB distribution map (**d**) pole figures (**e**) Inverse pole figure.

**Figure 15 materials-15-07708-f015:**
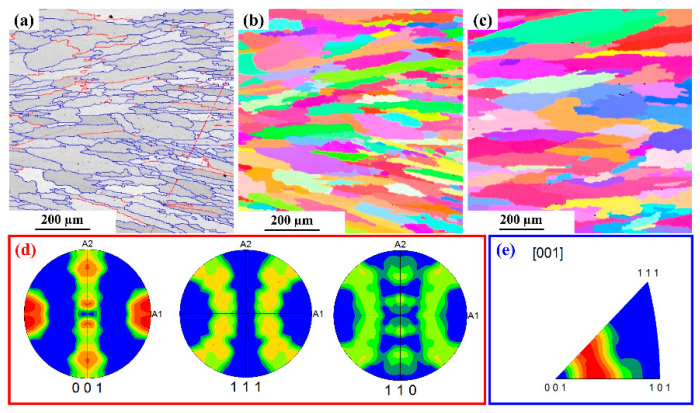
The Microstructure of the weld pool region (**a**) HAB and LAB distribution map (**b**) Inverse pole figure color map of weld pool region on the left side of the weld central line (**c**) Inverse pole figure color map of weld pool region on the right side of the weld central line. (**d**) Pole Figures (**e**) Inverse pole figures.

**Figure 16 materials-15-07708-f016:**
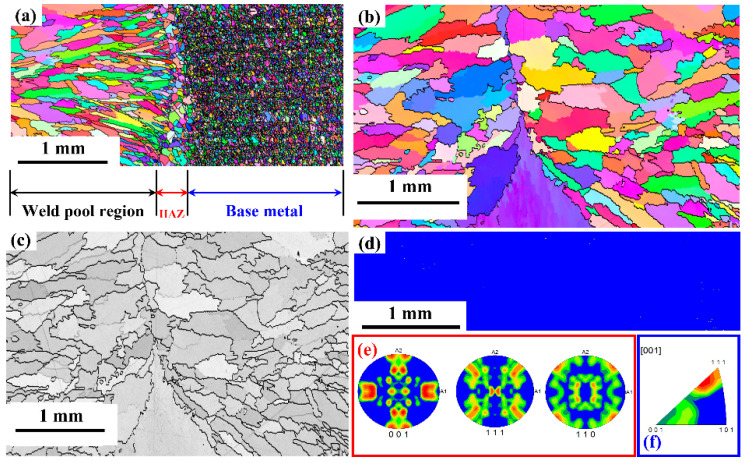
EBSD maps of complete L605 welds (**a**) Inverse pole figure map of the weld bead and base metal (**b**) Inver pole figure map of Weld pool region (**c**) Image quality map of weld pool region (**d**) Phase map (**e**) Pole figure of weld pool region (**f**) Inverse Pole figure of weld region.

**Figure 17 materials-15-07708-f017:**
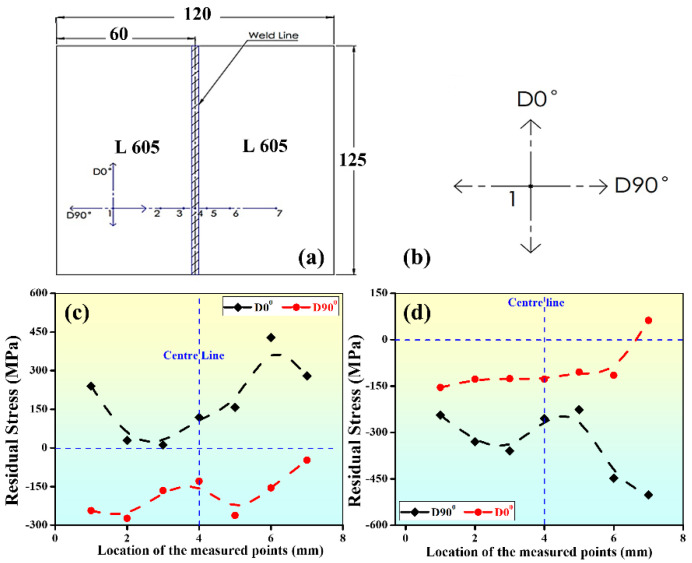
Residual stress measurement (**a**) Points on weld sample sketch, (point 4 represents the center of the weld bead) (**b**) Representation of D0° & D90° (**c**) Plot at D0° & D90° for weld sample with parameters 1800 W, 200 mm/min, (**d**) Plot at D0° & D90° for weld sample with parameters 1400 W, 600 mm/min.

**Figure 18 materials-15-07708-f018:**
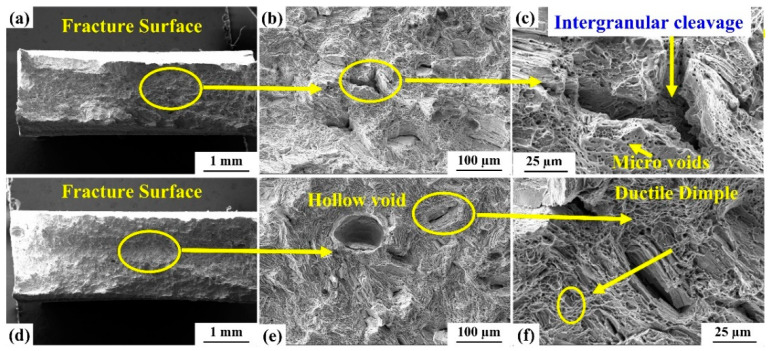
Fractography analysis of the specimens after conducting a tensile strength test with (**a**) high heat input parameters (exp.6), (**e**) low heat input parameters (exp. 9). (**b**) Lower magnifying image of fracture surface (exp 6) (**c**) Higher magnifying image of fracture surface (exp 6) (**d**) Overall Tensile fracture surface (exp 9) (**f**) Higher magnifying image of fracture surface (exp 9).

**Table 1 materials-15-07708-t001:** Chemical composition of L605 material.

Unit	Co	Cr	W	Ni	Fe	Mn	Si	C	S	P
Wt.%	Balance	20.00	15.00	10.00	3.00	1.50	0.40	0.10	0.030	0.040

**Table 2 materials-15-07708-t002:** The different input welding parameters used in the present study.

Parameter	Units	Level 1	Level 2	Level 3
Laser power	Watt (W)	1400	1600	1800
Welding speed	mm/min	200	400	600

**Table 3 materials-15-07708-t003:** Experiment table and the measured responses.

Run Order	Pt Type	Blocks	P	WS	W1	W2	H1	H2	Y
1	0	1	1600	400	4016.83	3807.56	139.51	145.33	492
2	1	1	1800	600	3051.86	2662.39	110.45	0	509
3	0	1	1600	400	3650.61	3859.88	156.95	122.07	481
4	1	1	1400	200	4964.36	4656.27	226.71	168.58	474
5	1	1	1800	200	5958.4	4447.00	610.37	1150.99	299
6	1	1	1400	600	2336.85	1790.43	122.07	87.20	532
7	0	1	1600	400	4016.83	3731.99	244.15	198.82	486
8	−1	2	1600	200	5981.65	5830.51	412.73	494.11	540
9	0	2	1600	400	3784.31	3749.43	168.58	133.70	506
10	−1	2	1400	400	3197.19	2371.73	198.82	129.40	495
11	0	2	1600	400	3877.32	3685.49	238.34	249.96	487
12	−1	2	1600	600	3121.62	2348.48	104.64	133.70	548
13	0	2	1600	400	3830.81	3929.63	267.4	191.83	495
14	−1	2	1800	400	4150.53	3865.69	325.53	395.29	409

P: Laser power (W), WS: Scanning Speed (mm/min), W1: Top width of weld bead, W2: Bottom width of weld bead H1: Weld bead height from the parent work surface, H2: Weld undercut bottom side.

**Table 4 materials-15-07708-t004:** Analysis of Variance for different responses.

	W_1_		W_2_		H_1_		H_2_		Y	
Source	**DF**	***p*-Value**	**DF**	***p*-Value**	**DF**	***p*-Value**	**DF**	***p*-Value**	**DF**	***p*-Value**
Linear	2	0.000	2	0.000	2	0.000	2	0.000	2	0.003
Square	2	0.001	2	0.029	2	0.137	2	0.017	2	0.026
2-Way Interaction	1	0.357	1	0.153	1	0.003	1	0.000	1	0.039
Lack-of-Fit	3	0.444	3	0.001	3	0.792	3	0.060	3	0.001
R-sq	98.11%		93.47%		92.97%		95.62%		84.97%	
R-sq(adj)	95.78%		89.39%		88.57%		92.89%		75.58%	

## Data Availability

The data may be made available on reasonable request to the authors.

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
