# Peer review of "Fiber Laser Welded Cobalt Super Alloy L605: Optimization of Weldability Characteristics"

_materials, 2022, doi:10.3390/ma15217708_

Round 1
Reviewer 1 Report
1. In the introduction, it is necessary to state things such as welding problems of cobalt-based superalloys, and the necessity of their welding. Also, the reasons for using laser welding, types of laser welding methods, and the reason for using fiber laser should be stated.
2. Why did you use the PT test to check for defects? Isn't the welding bed in the laser small for this test? So what about subsurface defects?
3. Show the numbers and figures related to lines 217 to 230, which are related to welding geometry, with a graph or table so that they can be compared better.
4. Many parts of the manuscript that appear in the results section should be included in the experiments section.
5. Microstructural investigations are not well discussed. Authors must pay most of each seed's share. Express the effect of laser parameters on microstructural changes. Explain the role of granule formation in mechanical and fracture properties. There is also no discussion about leaving. All numerical results should be presented in tables or graphs.
Overall, it is a good article, provided that it can be improved. Analyze more. You can get help from various references in this field. for example”10.1016/j.engfailanal.2022.106524” “10.1016/j.jmrt.2021.11.007”
Author Response
Comment 1. In the introduction, it is necessary to state things such as welding problems of cobalt-based superalloys, and the necessity of their welding. Also, the reasons for using laser welding, types of laser welding methods, and the reason for using fiber laser should be stated.
Different welding processes with high metal deposition rate like Gas Metal Arc Welding (GMAW), Gas Tungsten Arc Welding (GTAW) showed wide HAZ and welding defects in the joints, which restricts its uses in precision aerospace components [5]. Laser welding produces minimum defects and narrow HAZ which attracts its applications in various industries. Although there are several laser welding processes (using gas lasers and solid-state lasers), fiber laser welding is preferred over other processes due to its low maintenance cost and high reliability of the laser source.
Comment 2. Why did you use the PT test to check for defects? Isn't the welding bed in the laser small for this test? So what about subsurface defects?
The scales in the dye penetration and radiography test images are added for reference. Both the tests were conducted after the welding experiments and were carried out over the whole length of the weld bead. The dye penetration test provides the qualitative information of the surface defects, and the radiography test provides the internal or subsurface defects. Based on the results, tensile tests specimens were cut from the defects free zone of the weld bead.
Comment 3. Show the numbers and figures related to lines 217 to 230, which are related to welding geometry, with a graph or table so that they can be compared better.
The numbers and figures related to lines 217 to 230 have been explained elaborately. As we have followed the design of the experiment, the responses cannot be compared if given in the form of a table. However, the responses with values are given in table 3. The revised part of the manuscript in this regard is given below.
3.2.1 Weld bead geometry analysis
The weld bead geometry plays a vital role for the better strength of the joint. One of the typical weld bead geometries is shown in Fig. 3(c) for reference purpose and the measured values are presented in Table 3. Equation 2 to 6 represents the variation of geometrical parameters (responses) corresponding to the inputs which are explained with the help of the contour plots in Fig. 6. The width of the weld zone is reduced with the increase in scanning speed, this might be due to laser beam passes quickly over the weld line and hence less amount of base metal is melted, however, the weld width increases reasonably with the increase in laser power. With the increase of laser power, the bead width increases due to over melting of the base material (Fig. 6(b)). The bead width (W2) at lower side of the welding is found to be reduced with the increase in scanning speed, it may be due to the formation of keyhole in the welding process which leads to wider top and narrow bottom of the weld bead (Fig. 6(c)). The bead height and undercut are found to be maximum at higher laser power and lower scanning speed, at the appropriate combination of laser power 1600 W and scanning speed of ~500 mm/min the bead height and undercut are found to be minimum as shown in fig. 6(d, e). It shows the variation of W1, W2, H1, and H2 with respect to the laser beam power and welding speed. It has been observed that the minimum weld bead geometry (Table 3) with top width (W1): 2336.85 µm, Bottom width (W2): 1790.43 µm height (H1): 104.64 µm and undercut (H2): 0 µm were obtained at different parameter setting of laser power in the range of 1400 W to 1800 W and scanning speed 200 to 600 mm/min and the same is also confirmed from the contour plots shown in Fig. 6. The optimum (Fig. 9) values of the bead geometry can be obtained at parameters settings of laser power 1444.44 W and scanning speed 600 mm/min where top width (W1): 2565.84 µm, bottom width (W2): 1819.71 µm, height (H1): 122.39 µm and undercut (H2): 127.10. In view of the above, the welding parameters can be selected from the contour plots shown in Fig. 6 as per the requirement of the weld bead geometry, which is in line with the reference article [31].
Figure 6. Contour plots for weld bead geometry: (a) variation of tensile yield strength (Y) (b) variation of top width of weld bead (W1), (c) variation of bottom width of weld bead (W1), (d) variation of weld bead height (H1), (e) variation of undercut (H2), w.r.t input parameters.
Comment 4. Many parts of the manuscript that appear in the results section should be included in the experiments section.
The paper has been checked thoroughly and the required parts are shifted to experiment section.
Comment 5. Microstructural investigations are not well discussed. Authors must pay most of each seed's share. Express the effect of laser parameters on microstructural changes. Explain the role of granule formation in mechanical and fracture properties. There is also no discussion about leaving. All numerical results should be presented in tables or graphs.
The details of the microstructural investigation 1 presented in section 3.3.1. The numerical values of EDS analysis are presented in the Fig. 11(d) and Fig. 12(d).
Reviewer 2 Report
This work shows a detailed investigation on laser welded L605 alloy in terms of microstructure, hardness, tensile strength and their correlation. Some points can be considered before the final decision. 1. The article title cannot fully reflect the content and finings of the investigation, it is suggested to improve it; 2. The scale bar appears not proper according to the plate thickness with 2 mm, in some pictures; 3. The authors should remove the significant digits particularly for tensile properties, and also for other measurement; 4. This paper contains many pictures that indicates they lack a analysis, they can see the paper when studing this points, "on the fatigue performance of laser hybrid welded high Zn 7000 alloys for next generation railway components" and "the microstructure, mechanical and fatigue behaviors of MAG welded G20Mn5 cast steel"; 5. It is suggested to refine their findings, the conclusion shouldnot contains so many points, some are not; 6. The structure or organization should be again considered because it is not clear for us.
Author Response
The article title cannot fully reflect the content and findings of the investigation, it is suggested to improve it;
The title of the article has been revised to reflect the total content which is as follows: Fiber Laser Welded Cobalt Super Alloy L605: Optimization of Weldability Characteristics.
Comment 1. The scale bar appears not proper according to the plate thickness with 2 mm, in some pictures;
The same has been revised.
Comment 2. The authors should remove the significant digits particularly for tensile properties, and also for other measurement;
The insignificant digits for tensile properties and other measurements have been removed in the revised manuscript.
Comment 3. This paper contains many pictures that indicates they lack a analysis, they can see the paper when studing this points, "on the fatigue performance of laser hybrid welded high Zn 7000 alloys for next generation railway components" and "the microstructure, mechanical and fatigue behaviors of MAG welded G20Mn5 cast steel";
As per the review comments “on the fatigue performance of laser hybrid welded high Zn 7000 alloys for next generation railway components" and "the microstructure, mechanical and fatigue behaviors of MAG welded G20Mn5 cast steel"; has been studied and added the information acquired from the paper and cited as well.
Wu, S. C., Y. N. Hu, Hao Duan, C. Yu, and H. S. Jiao. "On the fatigue performance of laser hybrid welded high Zn 7000 alloys for next generation railway components." International Journal of Fatigue 91 (2016): 1-10.
Wu, S. C., Q. B. Qin, Y. N. Hu, R. Branco, C. H. Li, C. J. Williams, and W. H. Zhang. "The microstructure, mechanical, and fatigue behaviors of MAG welded G20Mn5 cast steel." Fatigue & Fracture of Engineering Materials & Structures 43, no. 5 (2020): 1051-1063.
Comment 4. It is suggested to refine their findings, the conclusion should not contains so many points, and some are not;
The conclusion has been refined as per the finding.
Comment 5. The structure or organization should be again considered because it is not clear for us.
The organization of the article has been checked and corrected wherever it is required.
Reviewer 3 Report
TITLE OF PAPER:
Fiber Laser Welded Cobalt Super Alloy L605: Weldability Characteristics.
REVIEWER’S COMMENTS :
The paper deals with mechanical and metallurgical properties of laser welded joints of cobalt superalloy under different beam power and welding speed. This paper is of an acceptable quality and can be published. However, some minor corrections are required to improve the paper as follows:
1. Results of tensile tests in Sub Section 3.2.3 should be deeply discussed by correlating strength with the weld microstructure or other parameters. In addition, Figure 8 should be accompanied by quantitative data on yield strength presented using Table or Graph.
2. In Abstract, the authors found an optimum input parameter which produced sound weld properties, i.e. P=1444,44 W and WS=600 mm/min. Please, add the value of P/WS which indicates heat input in Abstract.
3. The word of “%” for Table 1 is not clear whether weight percent (wt. %) or atomic percent (at. %). Please, give some explanations.
4. Please, mark various zones (Weld Metal, HAZ, Base Metal) in Figure 7.
5. The paper deals with Cobalt-based superalloy L605 which is widely used in high-temperature applications. However, the material properties reported in the paper do not represent actual operating condition (high temperature). Therefore, these need to be considered by the authors for future work.
Author Response
The paper deals with mechanical and metallurgical properties of laser welded joints of cobalt super alloy under different beam power and welding speed. This paper is of an acceptable quality and can be published. However, some minor corrections are required to improve the paper as follows:
Comment 1: Results of tensile tests in Sub Section 3.2.3 should be deeply discussed by correlating strength with the weld microstructure or other parameters. In addition, Figure 8 should be accompanied by quantitative data on yield strength presented using Table or Graph.
Sub section 3.2.3 has been revised and highlighted in the revised manuscript. “It was also observed that as the heat input supply during the welding is reduced, the cooling rate increases, and it results in the formation of finer grain size and therefore results in enhanced mechanical properties [35]. In this way, laser welding parameters at high input i.e., 1800 W laser power and 200 mm/min scanning speed resulted in low yield stress of 299 MPa as compared to the laser welding parameters at low heat input i.e., 1400 W laser power and 600 mm/min, where yield stress of 532 MPa was obtained.”. Moreover, yield strength data of all the samples are presented in the table 3.
Comment 2: In Abstract, the authors found an optimum input parameter which produced sound weld properties, i.e., P=1444.44 W and WS=600 mm/min. Please, add the value of P/WS which indicates heat input in Abstract.
P/WS value of the optimum parameter i.e., P=1444.44 W and WS=600 mm/min that indicates the line energy or heat input has been incorporated in the in the abstract section of the revised manuscript. (Line no- 27).
Comment 3: The word of “%” for Table 1 is not clear whether weight percent (wt. %) or atomic percent (at. %). Please, give some explanations.
Table 1 has been corrected where “%” is replaced by “wt. %” in the revised manuscript.
Comment 4: Please, mark various zones (Weld Metal, HAZ, Base Metal) in Figure 7.
Different zones in welded samples likes Fusion zone, HAZ, Base Metal zones have been marked in the Fig. 7 of the revised manuscript.
Comment 5: The paper deals with Cobalt-based superalloy L605 which is widely used in high-temperature applications. However, the material properties reported in the paper do not represent actual operating condition (high temperature). Therefore, these need to be considered by the authors for future work.
All experiments were conducted at room temperature and the material properties reported here also carried at room temperature. However, testing and characterization of high operating temperature will be considered as a part of future work. (Line No: 462-464).
Round 2
Reviewer 1 Report
The manuscript is well-edited and in my opinion, deserves to be released in Materials.